15

# Wildfire aerosols lofted by North American pyrocumulonimbus clouds: long-range transport and aerosol-cloud-radiative effects

Yan Wang<sup>1,2,3</sup>, Yujia Cao<sup>1,2,3</sup>, Haixiao Yu<sup>1,3</sup>, Shuo Wang<sup>1</sup>, Qiaoyun Hu<sup>4</sup>, Yuyang Chang<sup>4</sup>, Chun Zhao<sup>5</sup>, Zhengqiang Li<sup>6</sup>, Cheng Chen<sup>1,3</sup>

- <sup>1</sup> Anhui Institute of Optics and Fine Mechanics, Hefei Institutes of Physical Science, Chinese Academy of Sciences, Hefei 230031, China
- <sup>2</sup> Science Island Branch, Graduate School of University of Science and Technology of China, Hefei 230026, China
- <sup>3</sup>Anhui Province Key Laboratory of Optical Ouantitative Remote Sensing, Hefei Institutes of Physical Science, Chinese Academy of Sciences, Hefei 230031, China
- <sup>4</sup> Univ. Lille, CNRS, UMR 8518 LOA Laboratoire d'Optique Atmosphérique, F-59000 Lille, France
- <sup>5</sup> School of Earth and Space Sciences, University of Science and Technology of China, Hefei 230036, China
- <sup>6</sup> State Environment Protection Key Laboratory of Satellite Remote Sensing, Aerospace Information Research Institute, Chinese Academy of Sciences, Beijing 100101, China

Correspondence to: Cheng Chen (cheng.chen@aiofm.ac.cn)

Abstract. Extreme wildfires threaten health, air quality, and ecosystems. Despite extensive study of meteorological links, the feedback mechanisms by which fire weather influences the long-range smoke transport remain poorly understood. This study examines the transcontinental transport of smoke aerosols emitted by intense North American wildfires in August 2024. Our analysis reveals that pyrocumulonimbus clouds (PyroCbs) formed in extreme fires exhibit strong vertical convection, rapidly injecting large amounts of smoke aerosols into the upper troposphere and lower stratosphere. These lofted aerosols exhibit enhanced hydrophilicity at high altitudes, increasing the cloud condensation nuclei by a factor of 2-3. Therefore, the water cloud droplet effective radius decreases by approximately 1/3, and the cloud fraction increases from 0.01 to 0.64, promoting the development of optically thick, high-level clouds. PyroCb efficiently lifts aerosols, prolonging their residence time and enabling long-range transport through high-altitude winds. This process significantly affects regional and global radiation, with aerosols heating downwind Western Europe. Smoke aerosols produced consistent effects on radiative fluxes: they reduced longwave fluxes, while increased shortwave fluxes, resulting in net anomalies of +2.84 W m<sup>-2</sup> over fire sources and +3.16 W m<sup>-2</sup> in smoke-transported areas. Conversely, these aerosols over fire-source regions caused heterogeneous radiative responses, with net cooling anomalies of -2.5 W m<sup>-2</sup> along the US West Coast and +5.62 W m<sup>-2</sup> across North America. Our findings underscore the complex interplay between wildfires, smoke aerosols, and meteorology, forming a positive feedback loop that amplifies air pollution transport and radiative perturbations across continental scales.

#### 35 1 Introduction

Wildfires are increasingly recognized as major perturbations to the Earth's climate and atmospheric composition. They emit large quantities of trace gases and aerosols, including carbon monoxide (CO), black carbon (BC), organic carbon (OC), and various precursors of secondary pollutants, which strongly affect regional air quality, cloud formation, and radiative forcing (Andreae and Merlet, 2001; van der Werf et al., 2017). The atmospheric transport and vertical redistribution of wildfire plumes, particularly during intense events, can extend their influence far beyond the fire source region, contributing to hemispheric-scale smoke dispersion and stratospheric aerosol loading (Peterson et al., 2018). These fire-driven changes in aerosol-cloud-radiation interactions can result in significant perturbations to the top-of-atmosphere (TOA) energy balance (Khaykin et al., 2020a; Yu et al., 2006), yet the magnitude and structure of such effects remain poorly constrained.

Extensive observational and reanalysis studies have demonstrated that intense wildfires can rapidly lift smoke plumes through the PyroCb and inject them into the upper troposphere and even the stratosphere, thereby potentially affecting regional to cross-regional radiation budgets and circulation patterns (Fromm et al., 2010; Katich et al., 2023; Peterson et al., 2018). Comprehensive evidence from multi-platform satellite remote sensing and reanalysis data indicates that the stratospheric smoke load generated by extreme PyroCb events can be comparable to that of moderate volcanic eruptions and can sustain hemispheric-scale impacts for weeks to months (Doglioni et al., 2022; Katich et al., 2023). For cloud microphysics, smoke aerosols generally increase cloud droplet number, reduce cloud droplet effective radius, and enhance optical thickness (Twomey effect), thereby amplifying shortwave reflection, modulating precipitation efficiency and altering radiative forcing (Christensen et al., 2022; Conrick et al., 2021). Recent studies have also shown that wildfire smoke aerosols can influence the development and long-range transport of mid-latitude weather systems through radiation-dynamic coupling (Griffin et al., 2024; Wang et al., 2024). The 2017 Pacific Northwest PyroCb event, for instance, lofted an unprecedented amount of smoke into the lower stratosphere, with dramatic radiative implications (Kablick et al., 2020; Khaykin et al., 2020b; Lestrelin et al., 2021). Once injected, these smoke plumes can organize into anticyclonic structures that enable transcontinental transport and persistent stratospheric residence (Baars et al., 2019; Hu et al., 2019; Lestrelin et al., 2021).

The summer of 2024 wildfires in North America (particularly western Canada) were unusually active, resulting in significant transboundary smoke plume transport and PyroCb activity. This had a notable impact on air quality in Europe's downwind regions and on troposphere-stratosphere coupling processes (Damiano et al., 2024; Kolden et al., 2025). Due to the heatwave that swept across North America in early August, the number of fires and emissions in the Canadian Northwest region surged sharply in August. According to the estimates in August, Canada's carbon dioxide emissions exceeded 80 million tons, higher than the August of year 2023, the year with the highest carbon dioxide emissions according to the CAMS GFASv1.2 (Atmosphere Data Store, 2025; Kaiser et al., 2012). Satellite-based monitoring and model simulations indicate that on May 11-12, 2024, smoke from wildfires in British Columbia and Alberta produced maximum aerosol optical depth

https://doi.org/10.5194/egusphere-2025-5076 Preprint. Discussion started: 14 November 2025

© Author(s) 2025. CC BY 4.0 License.

80

(AOD) values of up to 5, moving eastward to affect Saskatchewan, Manitoba, Ontario, and bordering regions of the US, demonstrating that these fires released large amounts of pollutants and significantly impacted air quality (Filonchyk et al., 2025). Ground-based lidar and multi-source satellite joint inversion captured the structure and optical characteristics of smoke layers in the troposphere and lower stratosphere over central and southern Europe, supporting criteria for long-distance transport and vertical redistribution (Damiano et al., 2024). Concurrent satellite trace gas measurements and methodological studies provided a quantitative basis for characterizing the emission intensities and transport pathways of the 2024 North American wildfires, as well as observational and reanalysis evidence for their emission strength, transatlantic transport, and PyroCb-related convective injection. (Kolden et al., 2025; Skakun et al., 2024). Both mainstream and specialized media reported frequent "fire clouds" over North American wildfire sites in 2024, accompanied by long-range air quality impacts, emphasizing the climate relevance of PyroCb in the context of extreme fire years (Owens, 2024).

Previous studies have made significant progress in identifying the characteristics of wildfire-induced aerosols, describing the formation process of PyroCb, and quantifying radiative effects using multiple satellite and reanalysis data products (Balik et al., 2024; Jones et al., 2024; Kolden et al., 2025). However, existing studies are often limited in scope, frequently focusing on isolated case studies, single instrument datasets, or localized regions. These limitations hinder a comprehensive understanding of the complete causal chain from wildfire ignition, PyroCb generation, to subsequent smoke plume transport and aerosol-cloud-radiation effects. This study focuses on the wildfire event that occurred in the northern high latitudes of North America during summer 2024, and will use comprehensive observational and reanalysis datasets to characterize the widespread biomass burning, intense pyro-convection, and long-range transport of smoke aerosols, as well as the quantification of its associated aerosol-cloud-radiation effects.

## 2 Datasets and methodology

#### 2.1 Study region

To explore both the local and long-range atmospheric impacts of wildfires, this study focuses on two primary source regions and several downwind affected areas (Figure 1). The wildfire origin regions include the high-latitude regions of North America (NA, 55°-65°N, 130°-90°W) and the US West Coast (USW, 38°-48°N, 123°-113°W), where widespread biomass burning activity occurred in August 2024. In addition, we examine three long-range transport-affected receptor regions in Europe—namely Island of Great Britain (IGB, 50°-60°N, 10°W-2°E), Northern Scandinavia (NSCAN, 58°-71°N, 3°-20°E), and Western Europe (WE, 43°-53°N, 3°W-15°E), which are influenced by the advection of transcontinental smoke plumes. These areas were selected based on the spatial distribution of cloud albedo during periods of fire activity as shown by CERES SYN1deg-Month data, in combined with fire hotspots detected by the MODIS Terra.

Figure 1. Source regions and downwind areas analyzed to explore local and long-range wildfire impacts. Wildfire origin regions considered in this study are marked by blue rectangles: the high-latitude regions of North America (NA, 55°-65°N, 130°-90°W) and the US West Coast (USW, 38°-48°N, 123°-113°W). Regions impacted by long-range transported wildfire emissions are shown by gray rectangles: Island of Great Britain (IGB, 50°-60°N, 10°W-2°E), Northern Scandinavia (NSCAN, 58°-71°N, 3°-20°E), and Western Europe (WE, 43°-53°N, 3°W-15°E). The base map shows the 1°×1° cloud albedo effect from CERES\_SYN1deg-Month product, with red dots showing active fire detections over Northern American detected by the MODIS TERRA in August 2024.

#### 105 **2.2 Datasets**

A comprehensive suite of satellite observations, satellite-derived products, and meteorological reanalysis datasets was used in this study to investigate transcontinental aerosol transport driven by PyroCb during North American wildfire events in summer of 2024. The satellite observations comprised Terra/CERES, MODIS onboard Terra and Aqua, GOES-16/ABI, Suomi-NPP/OMPS, Suomi-NPP/VIIRS, Sentinel-5p/TROPOMI, EarthCARE/CPR and PACE/SPEXOne. To provide the meteorological context, we also incorporated the MERRA-2 reanalysis dataset. Key characteristics of these datasets, including spatial resolution, data source, and the variables employed, are summarized in Table 1.

Previous studies have mainly relied on burn area products such as MCD64A1 to estimate fire extent and emissions (Huang et al., 2023). However, such products are often limited by factors such as cloud cover, low temporal resolution, and overlap between Terra and Aqua satellite data, which may lead to overestimation or underestimation of the affected area (Giglio et al., 2018). Moreover, assessments of the radiative and dynamical effects of fire plumes have typically focused either on AOD retrievals or chemical transport model simulations, lacking consistent observational validation of vertical structure and TOA energy budget changes (Jaffe et al., 2020; Wilkins et al., 2022). In particular, convection-related vertical lifting mechanisms and cloud feedback triggered by fires are often insufficiently described in studies based solely on satellite observations.

To address these limitations, we employ a multi-sensor approach that combines MOD14A1 daily active fire products with the maximum fire radiative power (MaxFRP) classification to capture day-to-day variations in fire intensity and distribution (Freeborn et al., 2016; Randerson et al., 2012). This is complemented by MERRA-2 reanalysis for near-surface meteorology, aerosol composition, and upper-level circulation, including variables such as T2M (2-meter air temperature), RH (relative humidity), SLP (sea level pressure), 500/850 hPa wind fields, vertical velocity, and aerosol mass concentrations (Gelaro et al., 2017). To examine cloud responses and radiative forcing, we integrate geostationary GOES-16 BT/BTD data, Seninel-5p/TROPOMI CO column concentration, EarthCARE/CPR vertical reflectivity profiles, and CERES TOA flux observations. We further quantify fire-induced radiative effects using physical formulations of Climate Radiative Effect (CRE), net TOA perturbation (\( \Delta \) Net), and regional radiative forcing (RF), providing an observation-based evaluation of wildfire-climate coupling mechanisms.

Table 1. Overview of the satellite and meteorological reanalysis datasets used in this study: spatial resolution, data source, and selected variables.

| Data products                 | Resolution                | Source          | Variables                                                      |
|-------------------------------|---------------------------|-----------------|----------------------------------------------------------------|
| CERES SYN1deg                 | 1° × 1°                   | Terra/Aqua      | Observed TOA Fluxes and Cloud Parameters                       |
| -                             |                           | satellite       |                                                                |
| MOD14A1                       | 1 km                      | Terra satellite | Firemask and MaxFRP                                            |
| Mobi iii                      | ı Kili                    | /MODIS          |                                                                |
| MCD64A1                       | 500 m                     | Aqua and Terra  | Burned area                                                    |
| MCD04A1                       | 300 III                   | satellite/MODIS | Duriled area                                                   |
| MERRA-2                       | 0.5° × 0.625°             | MERRA           | Near-surface meteorological conditions; Wind and Geopotential  |
| MERRA-2                       | 0.5° × 0.625°             | reanalysis      | @500 hPa, 850 hPa and SLP; Aerosol mass concentrations (PM2.5) |
| OR_ABI-L2-MCMIPF-             | 2.1                       | COEC 16/ADI     | Data from multiple spectral bands of the ABI at Band 07        |
| M6_G16                        | 2 km                      | GOES-16/ABI     | (3.9 μm), and Band 13 (10.3 μm)                                |
| Suomi-NPP/VIIRS Imagery       | 750                       | Suomi-          | Data from multiple spectral bands of VIIRS at M3(0.64 µm),     |
| Resolution 6-Min L1B Swath    | 750 m                     | NPP/VIIRS       | M4(0.55 $\mu$ m) and M5(0.48 $\mu$ m)                          |
| 0.000.000.000.000.000.000.000 | <b>7</b> 0.1 <b>7</b> 0.1 | Suomi-          | ****                                                           |
| OMPS/NPP NMMIEAI-L2           | 50 km × 50 km             | NPP/OMSP        | UV Aerosol Index                                               |
| Seninel-5p/TROPOMI            |                           |                 |                                                                |
| L2_CO(Carbon                  | 5.5 km × 3.5 km           | Seninel-5p      | TROPOMI CO Total Column at SWIR 2.3 μm                         |
| Monoxide Total Column)        |                           | /TROPOMI        |                                                                |
| PACE_SPEX L2 RemoTAP          | ~2.5 km                   | PACE/SPEXone    | Retrieved Cloud Parameters                                     |
| ECA_JXCA_CPR_NOM_1B           | 500 m × 100 m             | EarthCARE/CPR   | Doppler velocity and radar reflectivity factor                 |

155

160

## 2.2.1 MODIS fire area detection and fire radiative power

Wildfire occurrence was characterized using the MODIS Collection 6.1 MOD14A1 daily active fire product onboard the Terra satellite, which detects thermal anomalies based on contextual thresholds applied to mid-infrared channels. MODIS is onboard both Terra and Aqua satellites, which operate in sun-synchronous orbits with a global revisit period of 1-2 days. It provides 36 spectral bands covering visible to thermal infrared regions and delivers products ranging from Level 1 radiance to Level 3 environmental applications (Fan et al., 2016; Justice et al., 2002; Masuoka et al., 2011).

While the burned area product MCD64A1 combines Terra and Aqua detections, it may introduce duplicate fire counts in overlapping swaths, potentially leading to overestimation in short-term fire quantification. Therefore, we use MOD14A1 for estimating daily fire activity, defined as Eq. (1):

145 Active Fire Area(Daily) = 
$$\sum_{\text{FireMask}\{7,8,9\}} A_{\text{pixel}} = N_{\text{fire pixels}} \times 0.25 \text{km}^2$$
, (1)

where FireMask values 1-9 correspond to low, nominal, and high-confidence fire detections. This provides a near-real-time spatial estimate of actively burning regions, unaffected by postfire cloud cover(Freeborn et al., 2014; Giglio et al., 2016). Fire intensity was assessed using the MaxFRP for each fire cluster, which is proportional to the combustion rate (Wooster et al., 2003). Based on MaxFRP, fire hotspots were further classified into low, moderate, and extreme-intensity categories.

## 150 2.2.2 MERRA-2 reanalysis meteorology and surface PM<sub>2.5</sub>

To characterize the meteorological environment during wildfire occurrence and plume transport, we used the MERRA-2 (Modern-Era Retrospective analysis for Research and Applications, Version 2) reanalysis dataset, produced by NASA's Global Modeling and Assimilation Office (Randles et al., 2017). MERRA-2 provides assimilated meteorological fields with  $0.5^{\circ} \times 0.625^{\circ}$  spatial resolution and hourly temporal frequency. Compared to earlier reanalysis dataset, MERRA-2 incorporates aerosol-meteorology interactions and assimilates satellite and ground-based remote sensed AOD, enabling more accurate assessment of thermodynamic and dynamic structures during extreme aerosol events (Buchard et al., 2017).

Near-surface meteorological conditions are evaluated using T2M, RH, and SLP/500/850 hPa wind fields, to diagnose fire-prone atmospheric settings. These variables have been widely used in previous studies of fire ignition and spread potential (Gutierrez et al., 2021; Jain et al., 2022). In addition, wind vectors and geopotential height fields at 850 hPa and 500 hPa are analyzed to characterize the large-scale circulation controlling aerosol transport. Vertical velocity is used to assess vertical transport and convection, particularly in regions with elevated fire radiative power and CO emissions.

To estimate aerosol loading near the surface, we extract MERRA-2 aerosol mass concentrations (kg kg<sup>-1</sup>) for BC, OC, SO<sub>4</sub>, dust, and sea salt (Buchard et al., 2017). These components were converted into PM<sub>2.5</sub> using mass-based scaling coefficients following Eq. (2):

(2)

as proposed in previous MERRA-2 PM studies (Provençal et al., 2017). The resulting spatial PM<sub>2.5</sub> maps provide quantitative insight into the near-surface health risk and smoke dispersion footprint (Ford et al., 2018).

## 170 2.2.3 EarthCARE Cloud Profiling Radar

The Earth Clouds, Aerosols and Radiation Explorer (EarthCARE) satellite is a collaborative mission between ESA and JAXA (Illingworth et al., 2015). Launched on May 28, 2024, EarthCARE aims to observe and characterize the vertical structure of clouds and aerosols, as well as their interactions with radiation. The satellite carries four primary instruments, including the Cloud Profiling Radar (CPR), which is the world's first spaceborne 94 GHz Doppler radar (Wehr et al., 2023).

The CPR provides vertical profiles of cloud reflectivity and vertical velocities, offering detailed insights into cloud and aerosol structures along the satellite's nadir track. The CPR Level-1B NOM\_1B product from the EarthCARE mission is used in this study. The NOM\_1B dataset provides calibrated radar reflectivity and Doppler velocity profiles along the satellite's nadir track, with a vertical resolution of approximately 500 m and a horizontal footprint of about 1 km. These measurements allow for detailed characterization of the vertical distribution of clouds and aerosol layers (Kollias et al., 2023; Mason et al., 2024).

## 2.2.4 Suomi-NPP OMPS UV aerosol index

The Suomi National Polar-orbiting Partnership (Suomi NPP) satellite carries multiple instruments that provide critical observations of the atmosphere and surface. Among these, the Visible Infrared Imaging Radiometer Suite (VIIRS) measures Earth's TOA reflectance at visible, near-infrared, SW infrared, and thermal infrared spectrum (VIS-NIR-SWIR-TIR) with a spatial resolution of 375 - 750 m at nadir and daily global coverage. This product enables the qualitative visualization of large-scale aerosol plumes, cloud systems, and surface features (Hillger et al., 2013). The Ozone Mapping and Profiler Suite (OMPS) Ultraviolet Aerosol Index (UVAI) product is also employed, which detects absorbing aerosols such as dust and smoke by quantifying the wavelength-dependent backscattered UV radiation. The OMPS UVAI has a nadir resolution of approximately 50 km × 50 km and is particularly suitable for tracking aerosol layers over bright surfaces such as clouds or deserts. Both datasets are distributed through the NASA Earth data portal (Chen et al., 2020).

# 2.2.5 GOES-16 Advanced Baseline Imager

The development and evolution of fire-induced deep convection were monitored using GOES-16/ABI (Advanced Baseline Imager) data (Schmit et al., 2005, 2019). GOES-16 is a geostationary satellite positioned over the Americas, providing full-disk imagery every 10-15 minutes in 16 spectral bands. It enables real-time tracking of convective processes at high temporal resolution, making it ideal for detecting PyroCb systems triggered by intense wildfires (Khlopenkov et al., 2021; Peterson et al., 2015).

## 2.2.6 Sentinel-5p/TROPOMI columnar CO

The Sentinel-5 Precursor (Sentinel-5P) mission, part of the Copernicus program, carries the Tropospheric Monitoring Instrument (TROPOMI) designed for global monitoring of atmospheric composition (Veefkind et al., 2012). In this study, the Level-2 carbon monoxide total column product (L2\_CO) was employed. This dataset is retrieved from measurements in SWIR band at 2.3 µm, which is sensitive to CO in the lower troposphere. The product provides total column CO concentrations with a nadir spatial resolution of 5.5 km×3.5 km, enabling the detection of both anthropogenic and biomass burning emissions (Landgraf et al., 2016; Torres et al., 2020)).

## 2.2.7 PACE/SPEXone derived aerosol and CCN

SPEXone is a multi-angle spectropolarimeter launched aboard NASA's PACE mission in 2024 (Werdell et al., 2019). To analyze aerosol microphysics and cloud interaction (van Diedenhoven et al., 2022), we employed SPEXone aerosol measurements (Hasekamp et al., 2019). It provides retrievals of AOD, fine- and coarse-mode volume fraction (Fu et al., 2025), effective particle radius (r<sub>eff</sub>), and cloud condensation nuclei (CCN) proxy indicators (Mishchenko et al., 2004). SPEXOne has high angular and spectral resolution that enable advanced retrievals of aerosol scattering phase function and absorption features. Time series of these derived parameters were extracted over the fire-affected region to examine changes in aerosol load and composition during the event. Other commonly used parameters to indicate the characteristic size and width of a size distribution are the effective radius of a single particle group and the effective radius of multiple particle groups (McFarquhar and Heymsfield, 1998; Stamnes et al., 2018), formally defined as Eq. (3)-(4), respectively:

$$r_{eff,i} = \frac{\int r^3 n(r) dr}{\int r^2 n(r) dr},$$
(3)

$$r_{\text{eff}} = \frac{\sum_{i} V_{i} r_{\text{eff},i}}{\sum_{i} V_{i}}, \tag{4}$$

## 2.3 Methodology

## 2.3.1 NOAA HYSPLIT Trajectory

To simulate long-range aerosol transport, we conducted forward trajectory analysis using the NOAA HYSPLIT (Hybrid Single Particle Lagrangian Integrated Trajectory) model (Rolph et al., 2017; Stein et al., 2015). Trajectories were initialized from regions of high fire intensity or UVAI concentration, and run for up to 72 hours using Global Data Assimilation System (GDAS) meteorological fields. This approach helps trace the evolution of smoke plumes and connect remote sensing observations of aerosols to their source regions, a method widely adopted in wildfire smoke studies (Colarco et al., 2004; Damoah et al., 2004; Dirksen et al., 2011; Yu et al., 2019).

## 2.3.2 TOA radiative forcing and cloud effect evaluation

To evaluate the TOA radiative impacts associated with wildfire-driven aerosol and cloud changes, we used the Clouds and the Earth's Radiant Energy System (CERES) Syn1Deg daily data product (Wielicki et al., 1996). CERES provides broadband shortwave (SW) and longwave (LW) radiative fluxes under both all-sky and clear-sky conditions, with global coverage and a spatial resolution of 1° × 1° (Doelling et al., 2013; Loeb et al., 2001). This allows the quantification of both the direct and indirect radiative effects of aerosols and cloud modifications induced by fire plumes (Loeb et al., 2018; Rutan et al., 2015a).

CRE was computed separately for SW, LW (Loeb and Schuster, 2008; Ramanathan et al., 1989), and  $\Delta$ Net components using the following Eq. (5):

$$CRE = (Rad_{all-sky, X} - Rad_{clear-sky, X}) \times F_{cloud},$$
(5)

where X includes SW, LW and net, Rad<sub>all</sub> and Rad<sub>clear</sub> represent the all-sky and clear-sky radiative fluxes (Doelling et al., 2013; Rutan et al., 2015b), and F<sub>cloud</sub> is the water cloud fraction obtained from CERES. This formulation isolates the net cooling or warming influence of clouds at TOA and highlights fire-induced cloud changes that may enhance or suppress regional radiation budgets (Liu et al., 2020; Rosenfeld et al., 2008). Using a dynamic cloud fraction in the weighting factor accounts for partial cloud cover and ensures more realistic representation of spatially variable cloud effects, particularly under active fire convection conditions (Liu et al., 2025; Loeb et al., 2009; Martins and Silva Dias, 2009).

To capture the short-term fire-induced radiative anomaly, we calculated the net TOA radiative perturbation ( $\Delta$ Net) using the following Eq. (6):

$$\Delta \text{Net} = -[(SW_{\text{Wildfires}} - SW_{\text{mean}}) + (LW_{\text{Wildfires}} - LW_{\text{mean}})], \qquad (6)$$

This equation represents the deviation in net radiative flux from climatological baseline conditions, where climatological values are constructed from a multi-year mean (2000-2020) for the same calendar period. The negative sign indicates net energy loss from the system. This approach is advantageous because it directly quantifies the net radiative effect of the fire event, without requiring assumptions about individual aerosol-cloud-radiation interactions. It has been successfully used in large-scale fire studies to attribute radiative changes to wildfire smoke and its interaction with meteorology (Liu et al., 2022, 2020; Mallet et al., 2020, 2021; Vaezi et al., 2025).

To assess the spatially aggregated radiative forcing, we calculated the area-weighted mean forcing over the plume-affected region using the following Eq. (7):

$$RF_{region} = \frac{1}{A} \int (Net_{pert} - Net_{mean}) dA, \qquad (7)$$

where A is the area of integration, and denote the perturbed and reference net TOA fluxes, respectively. This framework enables a quantitative assessment of regional energy imbalances driven by smoke-laden clouds and associated aerosol-radiation interactions. By capturing the combined influences of direct aerosol forcing, cloud adjustments, and dynamical feedbacks within a single observable metric, it provides a powerful constraint on climate sensitivity and feedback processes.(Climate Change 2021 - The Physical Science Basis, 2025; Anthropogenic and Natural Radiative Forcing — IPCC, 2025; Tosca et al., 2013; Wang et al., 2007)

## 3 Results

The 2024 North America wildfire season exhibited unprecedented characteristics compared to previous events, providing a critical case study for understanding the aerosol-cloud-radiation interactions caused by wildfires. Unlike the traditional wildfire season that typically occurs in summer, the 2024 wildfire season began unusually early and lasted longer than usual, partly due to the reburning of numerous "zombie fires" from the previous year (Scholten et al., 2021). The area burned and associated carbon emissions reached record highs, though second only to 2023 (Kolden et al., 2025). However, the fires were more concentrated in western Canada (Northwest Territories, British Columbia) and spread more rapidly, highlighting the intensification of fire regimes in Arctic and temperate ecosystems (Filonchyk et al., 2025). Additionally, wildfire smoke spread far beyond the source regions, severely worsening air quality in major U.S. cities and even European regions, thereby exacerbating public health risks across the continent and further highlighting the growing socioeconomic costs of wildfire disasters.

## 3.1 Weather-scale changes in wildfires and meteorology during extreme fire periods in typical fire regions

Figure 2 presents an integrated analysis of near-surface meteorological conditions, PM<sub>2.5</sub> pollution and fire-induced BC AOD over North America in August 2024. As shown in Figure 2a, in the fire-intensive region [blue rectangle in (Figure 2d)], the temporal evolution of 2-meter temperature (T2M), relative humidity (RH), vertical speed (VS) and 2-meter wind speed (WS), along with satellite-derived active fire area, reveals a distinct peak in fire activity between August 8-14. During the fully developed fire stage (August 4-10), the T2M significantly increased by 1.45 °C, the near-surface wind speeds generally strengthened by 0.5 m s<sup>-1</sup>, relative humidity sharply decreased around the fire area, and strong vertical uplift occurred. Meanwhile, T2M gradually recovered while relative humidity rose incrementally by 4.6 %. The smoke plume was primarily transported by background wind advection, exhibiting long-range horizontal dispersion and transport rather than intense vertical uplift, suggesting a meteorological environment conducive to wildfire ignition and spread. From Figure 2b, the monthly mean PM<sub>2.5</sub> concentration indicate significant pollution accumulation over the western Canadian boreal forest and parts of the Pacific Northwest. These fire pots correspond spatially to high fire activity and suggest persistent smoke emission and stagnant atmospheric conditions favourable for pollutant retention.

Figure 2. Spatiotemporal Distribution of Near-Surface Meteorological Variables, PM<sub>2.5</sub> Pollution, and Fire-Driven BC AOD over the Northern America in August 2024. (a) Daily satellite-detected burned area, T2M, RH, Vertical speed (Pa S<sup>-1</sup>) and 2-M wind speed (WS) from MERRA-2 in August 2024 in the fire-intensive region [blue rectangle in (Figure 2d)]. (b) Monthly Mean PM<sub>2.5</sub> concentrations during August 2024 in the Northern America. (c) Cross section of Firepots-increased AOD (contour lines) and BC surface concentration (shades) changes. Blue and orange solid dots represent small and medium fires, red crosses represent large fires, and red solid lines and enclosed areas represent high AOD areas. (d) Anomalies of Fire-Originated BC AOD (Red solid lines show the increase of AOD anomaly) effect of smoke aerosol.

Figure 2c captures the spatial structure of fire-enhanced BC aerosol concentrations and associated AOD anomalies on August 10, a peak fire day. MaxFRP values, particularly over central and northern British Columbia, are collocated with elevated BC surface concentrations (>25 μg m<sup>-3</sup>) and enhanced AOD contours, indicating both strong emission and effective vertical mixing of fire plumes. The spatial alignment between medium-to-large fire clusters and downwind aerosol plumes highlights the role of mesoscale transport mechanisms. Lastly, Figure 2d depicts the monthly anomalies in fire-originated BC and AOD, illustrating sustained deviations from climatological baselines across high-latitude regions. The most prominent anomalies are centred over northwest Canada, where fire-driven aerosol burdens are estimated to have significantly exceeded

the multi-year mean. The presence of positive AOD anomalies (>0.3) in these regions supports the conclusion that wildfire emissions in August 2024 contributed notably to the regional aerosol loading and radiative forcing.

Figure 3. Satellite Observations of Wildfire Smoke and Vertical Aerosol-Cloud Structure on August 13 and 17, 2024. (a, b) Suomi-NPP/VIIRS true-color imagery overlaid with Suomi-NPP/OMPS UV Aerosol Index on August 13 and 17, 2024, respectively. The black line indicates the EarthCARE satellite ground track, with the CPR reflectivity segment highlighted (note: width not to scale). Red shading indicates all TERRA and AQUA MODIS fire detections for the preceding 24h at native resolution. Panel (a) covers 10:00:25-20:09:22 UTC; panel (b) covers 11:47:50-21:05:06 UTC, both from east to west. (c, d) Radar reflectivity factor from EarthCARE/CPR along the satellite track on August 13 at 08:53:29 UTC and on August 17 at 15:06:57 UTC, respectively, showing the vertical structure of aerosol and cloud. Gray dashed lines show the boundary between the troposphere and the stratosphere.

To further assess the vertical structure of smoke-laden cloud systems, we utilized radar reflectivity profiles from the CPR onboard the EarthCARE satellite. As can be seen from Figure 3, CPR overpasses coinciding with OMPS UV Aerosol Index (UVAI) maxima allowed cross-examination of vertical layering in fire-related clouds and plumes (Fromm et al., 2005; Peterson et al., 2018). Figure 3a shows the spatial distribution of the UVAI high-value area across a large region of Canada on August 13. Combined with CPR radar echo results from the adjacent period (Figure 3c), the vertical echo profile reveals typical characteristics of deep convective clouds. In the middle and lower layers, there are precipitation particles or mixed-phase regions, while the upper layer has developed a thick ice anvil structure. Its morphology is similar to the deep convection and upper-level ice clouds commonly observed in PyroCb event. Figure 3b shows that the high-value UVAI zones on August 17 were distributed over the wildfire source areas in Canada and the eastern Atlantic region, indicating that wildfire smoke plumes had been transported to the western coastal waters of the European continent. Figure 3d shows the CPR radar echo results, indicating that the cloud body is primarily composed of high-altitude small ice crystals and snow

grains, with a lack of low-altitude precipitation echoes. This structure is typical of PyroCb clouds during the decay phase of their transport, when only high-altitude ice crystals remain, between 45° N and 65° N. When this phenomenon aligns with the smoke aerosol transport process, it further indicates that it is a fire-driven PyroCb ice cap remnant.

To quantitatively track the life cycle of thunderclouds, we used two key thermal infrared indicators from GOES-16/ABI (Figure 4a-b): the 11 µm brightness temperature (band 13), which is sensitive to the cooling characteristics of convective cloud-top updrafts; and the brightness temperature difference (BTD) between 4 µm and 11 µm, which effectively distinguishes high-temperature fire surfaces from cooler cloud layers. (Peterson et al., 2017). This parameter combination has been widely used to identify smoke aerosols exceeding cloud tops and fire-convection interactions in mid-latitude and northern hemisphere fire areas (Apke et al., 2018; Fromm et al., 2005). As shown in Figure 4, thermal infrared brightness temperature data acquired by the GOES-16 satellite's ABI spectrometer at the 11 µm wavelength band reveal supercooled cloud tops appearing as grayish-white in the image. This indicates the presence of deep convective cloud tops associated with PyroCb. Darker colors show lower cloud top temperatures, indicating higher and thicker cloud layers. The 4-11 µm band difference (BTD) identifies regions of enhanced reflectivity where PyroCb occurs, indicating smaller cloud top particles, which is typically associated with microphysical changes induced by smoke aerosols (colored regions in Figure 4). Satellite observations on August 13 (Figure 4a-b), reveal two blue-green bright areas over the northwestern Rocky Mountains of the United States and central-southern Canada, illustrating the spatial evolution and microphysical characteristics of the smoke aerosol plume during its slow eastward advection. The sustained increase in albedo coupled with relatively low infrared brightness temperatures indicates the ongoing vertical development and long-range transport of smoke aerosols within the upper troposphere. This evolution reflects both the convective intensity of the initial plume and the role of weather-scale dynamics in promoting cross-boundary aerosol diffusion.

Based on the satellite observations in Figure 4a-b, the specific locations where cumulonimbus clouds formed were identified as the overlapping areas between the black regions in the grayscale shaded map and the blue-green highlighted regions. Figure 4c-d displays the NOAA HYSPLIT forward trajectories of air masses originating from two cumulonimbus cloud regions: Region 1 at 44°N, 119°W, and Region 2 at 54°N, 109°W. For Region 1, air parcels initialized at 3 km were significantly lofted to altitudes approaching 9 km, indicating strong vertical transport likely induced by PyroCb activity. Similarly, in Region 2, both 3 km and 5 km air parcels were rapidly lifted to near 9 km altitude. By 18:00 UTC on 15 August, partial overlap of the two air mass trajectories occurred, suggesting convergence and coordinated long-range transport. The merged air parcels subsequently traversed the North Atlantic and reached Northern and Western Europe. These results highlight the role of PyroCb events in injecting wildfire smoke aerosols to the upper troposphere or even lower stratosphere, enabling fast, transcontinental transport that ultimately influences atmospheric composition over Europe.

**Figure 4. Cloud-Top Evolution and Smoke Transport Pathways of the North American PyroCb Event.** (a-b) North American PyroCb Event on 13 August 2024: Cloud-Top Structure, and PyroCb Cloud Activity. c-d, NOAA HYSPLIT Forward trajectories of Air Parcels Originating on August 13, 2024, and Reaching the Smoke Conveyor Region. (c) Region 1 at 44°N, 119°W, with trajectories initialized at 12:00 UTC on August 9 and arriving at the target region by 00:00 UTC on August 14, at three altitude levels: 4 km, 5.5 km, and 6 km. (d) Region 2 at 54°N, 109°W, with trajectories initialized at the same time but arriving by 00:00 UTC on August 19, at altitudes of 5.5 km, 6 km, and 7 km.

#### 3.2 Transcontinental transport of Canadian wildfire smoke

To analyse the key meteorological scale characteristics influencing the transport of wildfire smoke in North America, Figure 5 presents wind fields, wind speeds, and geopotential elevation at SLP, 850 hPa and 500 hPa levels. This integrated display reveals the background meteorological systems governing both the initial fire development area and the subsequent smoke

transport. At SLP, weak surface winds and low-pressure systems over the wildfire regions in both the USW and NA favor the persistence of fire activity, while the surface cyclonic circulations over the North Atlantic facilitate the initial dispersion of smoke plumes. At 850 hPa, enhanced southwesterly winds and low-level jets promote horizontal advection toward the east. Meanwhile, upper-level troughs and strong westerlies at 500 hPa facilitate vertical lifting and long-range transport of smoke aerosols across the North Atlantic. These multilevel dynamical structures jointly contribute to efficient lofting and transboundary movement of wildfire emissions.

Figure 5. Horizontal distributions of wind vectors (arrows), wind speed (shading, m/s), and geopotential height (contours, gpm). (a,b) SLP, (c,d) 850 hPa, and (e,f) 500 hPa over the North American region on 13 August (left column) and 17 August (right column) 2024.

On August 13 (Figures 4a-b), fire convective activity intensified, forming pyrocumulus clouds over the fire area. Combined with Figure 6a, a significant vertical rise in CO was observed, with enhanced concentration extending to approximately 600 hPa. Correspondingly, the vertical pressure-velocity field exhibited pronounced convective anomalies, with vertical pressure velocity less than -0.4 Pa s<sup>-1</sup> (indicating strong upward motion) directly above the fire center. This coupling of intense ground emissions with vertical transport reflects the rapid injection of fire-generated pollutants into the middle troposphere via convective uplift pathways.

In contrast, during the long-range transport phase on August 17 (Figure 6b), sporadic fires persisted across NA but diminished in intensity. Nevertheless, ascending air currents ( $\omega$  < -0.5 Pa s<sup>-1</sup>) remained dominant over multiple regions, sustaining the uplift of residual smoke plumes. CO profiles revealed extensive latitudinal distribution of pollutant concentrations, extending from approximately 120°W to 0°W, with vertical distribution reaching up to 550 hPa. This indicates efficient transatlantic transport of smoke aerosols, with background concentrations remaining persistently elevated due to weather-scale lifting and residual convective structures.

Figure 6. Vertical composite cross-sections of MERRA-2 Pressure-Level CO concentration and vertical pressure velocity (in 100 Pa/s) anomalies centered on high Column CO areas. The cross-sections are taken along the red line shown in Figure 5: (a) August 13 correspond to wildfire impacts and PyroCb development, while (b) August 17 represent the long-range transport of smoke aerosols.

## 4 Quantification of smoke induced aerosol-cloud-radiation effects

## 4.1 Influence of smoke aerosols on cloud formation

To assess the influence of smoke aerosols from intense wildfires on cloud microphysical properties, we analyzed CERES SYN1deg retrieved cloud parameters during and after the PyroCb events. The focus was on three variables: cloud-top height change ( $\Delta$ CTH), cloud optical depth change ( $\Delta$ COD), and cloud effective radius change ( $\Delta$ CER), which reflect changes in cloud vertical and microphysical structure.

400

395

https://doi.org/10.5194/egusphere-2025-5076 Preprint. Discussion started: 14 November 2025

© Author(s) 2025. CC BY 4.0 License.

405

420

EGUsphere Preprint repository

As shown in Figure 7, significant anomalies in cloud properties were observed over the wildfire source regions on August 13 and 15, 2024, coinciding with intense PyroCb development. The  $\Delta$ CTH exhibited marked increases of up to  $\sim$ 5 km above climatological norms, with local maxima exceeding 8km on both days (Figures 7a-b), indicative of vigorous deep convection triggered by intense fire-driven updrafts. The  $\Delta$ COD also showed substantial enhancement. On August 13,  $\Delta$ COD increased by approximately 25 (Figure 7e) in the core PyroCb regions (Figure 4a-b). On August 15, the  $\Delta$ COD anomaly region shifted eastward and expanded to nearly three times its previous extent (Figure 7f), reflecting enhanced  $\Delta$ COD along the smoke-affected convective belt.

ΔCER responded heterogeneously. On August 13, ΔCER decreased by up to 3 μm over the fire source region (Figure 7i), consistent with the Twomey effect, wherein elevated aerosol loading leads to more numerous but smaller droplets. Interestingly, an adjacent area downwind showed increased ΔCER, possibly due to secondary droplet growth or aerosol aging during transport. On August 15, ΔCER showed widespread suppression across the affected NA region, with anomalies reaching of -2 μm to -5 μm (Figure 7j), indicating persistent microphysical modification under elevated smoke aerosol concentrations. These results demonstrate that fire-induced aerosol loading not only intensified convective development but also alters cloud microphysical processes during the active burning period.

Figure 7 also compares the cloud properties during the downwind transport phase of the wildfire plume on August 17 and 19, 2024. Elevated  $\Delta$ CTH persisted along the long-range smoke transport pathways, with anomaly values exceeding +5 km in several regions (Figures 7c-d), suggesting sustained mid-to-upper tropospheric cloud development influenced by advected smoke layers. The  $\Delta$ COD remained anomalously high, with increases reaching ~20 units in the downstream regions, particularly over WE and the IGB (Figures 7g-h). This indicates a continued enhancement of  $\Delta$ COD even several days after the peak PyroCb events.

ΔCER also displayed consistent suppression in smoke-affected regions (Figures 7k-l), supporting the presence of aged wildfire aerosols influencing cloud microphysical properties via long-range interactions. These results highlight the prolonged impact of wildfire smoke on cloud systems far from the fire origin, reinforcing the hypothesis that biomass burning aerosols can induce remote cloud microphysical perturbations and potentially alter the regional radiation budget and precipitation efficiency across transcontinental scales. Such alterations may contribute to net radiative forcing anomalies at the regional scale.

Figure 7. Anomalies of Retrieved Cloud Properties change Relative to Multi-Year Mean on August 13, 15, 17 and 19, 2024 from CERES SYN1deg Product. Each row presents spatial anomalies of (a-d)  $\Delta$ CTH (km) (e-h)  $\Delta$ COD (km), and (i-l)  $\Delta$ CER ( $\mu$ m), computed as the difference between observed values during the PyroCb event and multi-year climatology. The cloud property inversion values correspond to observations on August 13, 15, 17, and 19, respectively, illustrating cloud disturbances induced by intense wildfire-driven convection and the evolution of cloud microphysical properties triggered by wind-transported wildfire aerosols.

Figure 8. Impact of the North American wildfire on Aerosol Properties and Compositional Properties (August 1-14, 2024) retrieved from PACE/SPEXone with RemoTAP algorithm. Daily mean (lines) and standard deviation (shaded) of (a) AOD, Fine- and Coarse-mode AOD, and AAOD. (b) BC and OC volume fractions. (c) Volume density for three particle models. (d) Satellite-retrieved volume fractions of major aerosol components.

During the North American wildfire episode from August 1 to 14, 2024, a significant increase in aerosol loading and compositional changes were observed (Figure 8). As shown in Figure 8a, AOD exhibited a marked rise, with both fine- and coarse-mode components contributing to the enhancement. The fine-mode AOD increase suggests a dominance of submicron particles typically associated with biomass burning. Concurrently, the absorption aerosol optical depth (AAOD) also rose, indicating the presence of light-absorbing aerosols such as BC. Figure 8b highlights the temporal evolution of aerosol composition, where volume fractions of BC and OC showed substantial increases during the peak wildfire days (August 6-11). The elevated BC fraction suggests strong emissions from incomplete combustion, while high OC levels reflect the organic-rich nature of smoke plumes. These changes are consistent with the satellite-retrieved aerosol component volume fractions (Figure 8c-d), which further confirming that carbonaceous aerosols caused the increase in AOD and AAOD during this event. Overall, these results point to the significant perturbation of the atmospheric composition caused by wildfire emissions, with potential implications for regional radiative forcing, air quality, and aerosol-cloud interactions.

Figure 9 illustrates the temporal evolution of key aerosol-cloud interaction variables during the North American wildfire episode from August 1 to 14, 2024. Figure 9a shows that CCN concentrations increased significantly across all supersaturation levels at different SS levels (SS = 0.1%, 0.5%, 0.7%), particularly between August 6 and 11. The most pronounced enhancement was observed at SS = 0.5%, suggesting a substantial rise in accumulation-mode aerosols capable of activating into cloud droplets. This increase is consistent with the elevated levels of fine-mode aerosol and carbonaceous species (Figure 8d), which are known to be efficient CCN. In Figure 9c, the cloud fraction exhibits a significant and sustained increase following the decay stage of the wildfire, during the August 11-14, indicating a pronounced cloud response to enhanced aerosol loading. This increase closely follows the sharp rise in CCN concentrations shown in Figure 9a, particularly during August 6-11, when wildfire emissions were most intense. Concurrently, Figure 9b shows a clear decrease in aerosol effective radius, reflecting the injection of large numbers of small, fine-mode particles from biomass burning. This shift in aerosol size characteristics directly influences cloud microphysical properties, as smaller particles can lead to the formation of more numerous but smaller cloud droplets under a given liquid water content.

The simultaneous enhancement in CCN and reduction in particle size supports the occurrence of the Twomey effect, where in an increased number of smaller CCN leads to the formation of more numerous, smaller cloud droplets. This microphysical adjustment increases COD and cloud albedo, resulting in more reflective clouds. The observed rise in cloud fraction further suggests that aerosol-cloud interactions extended beyond microphysical properties to influence cloud microphysical characteristics, potentially increasing cloud cover through cloud lifetime or formation effects.

Figure 9. Influence of Wildfire-induced Aerosol Loading on CCN and Cloud Properties in North America (August 1-14, 2024).

Daily mean (lines) and standard deviation (shaded) of (a) CCN at SS=0.1%, 0.5% and 0.7%, (b) Aerosol Effective Radius, (c) Cloud fraction. Gray shading marks wildfire activity; red shading marks PyroCb development.

## 4.2 Influence of smoke aerosols on TOA radiation

Figures 10 and 11 illustrate the spatial patterns of SW, LW, and net in TOA radiative fluxes under both all-sky and clear-sky conditions, including the climatological mean, wildfire-period mean, and their anomalies, computed as anomalies relative to the multi-year climatological mean for August obtained from CERES SYN1deg.

Under clear-sky conditions, all regions show negative outgoing SW anomalies, ranging from -2.21 to -7.07 W m<sup>-2</sup>, indicating a reduction in reflected solar radiation and enhanced atmospheric absorption by smoke aerosols. The strongest negative anomalies occur over WE (-7.07  $\pm$  0.22 W m<sup>-2</sup>) and NA (-4.48  $\pm$  0.08 W m<sup>-2</sup>), consistent with heavy smoke loading in the source regions. Under all-sky conditions, the SW anomalies vary in sign, reflecting the competing effects of smoke absorption and aerosol-cloud interactions. Negative anomalies over USW (-0.47  $\pm$  0.28 W m<sup>-2</sup>) and NSCAN (-3.54  $\pm$  0.29 W m<sup>-2</sup>) suggest that absorbing smoke layers located above clouds suppressed cloud-top reflection. In contrast, positive anomalies in NA (+3.58  $\pm$  0.28 W m<sup>-2</sup>), WE (+3.12  $\pm$  0.36 W m<sup>-2</sup>), and IGB (+0.35  $\pm$  0.38 W m<sup>-2</sup>) indicate enhanced reflection caused by smoke-induced cloud brightening during long-range transport. These results demonstrate that direct aerosol absorption dominates in the source regions, while aerosol-cloud interactions modulate the SW radiative response in transported plumes.

The LW anomalies exhibit distinct spatial patterns under different sky conditions. Under clear-sky conditions, LW anomalies are generally positive (0.01-4.95 W m<sup>-2</sup>), indicating slightly enhanced outgoing thermal radiation. This enhancement may result from surface or lower-atmospheric warming induced by smoke absorption, which increases the emission of LW radiation to space. In contrast, under all-sky conditions, LW anomalies vary between negative and positive values (-8.15 to +3.30 W m<sup>-2</sup>), reflecting more complex aerosol-cloud-radiation interactions. Negative anomalies over NA, mainly observed over dense smoke and cloud-overlapping regions, suggest reduced emission of thermal radiation due to cloud-top cooling and the presence of optically thick smoke layers. Positive anomalies in other regions likely correspond to areas with less cloud cover or enhanced lower-tropospheric heating. In the USW region (+3.30 ± 0.22 W m<sup>-2</sup>), wildfire smoke layers suppress convection and inhibit cloud formation, causing existing clouds to exhibit lower cloud tops (Figure 7a-d).

The net anomalies, defined as downward positive, reflect the combined changes of SW and LW fluxes. The observed regional pattern is not uniform. In the wildfire-origin regions the net anomalies are mixed: USW shows negative net anomalies under both conditions, while NA shows positive net anomalies. In the long-range transport regions, the net anomalies are consistently positive or close Climatology (clear-sky: -0.18 to 3.57 W m<sup>-2</sup>) (all-sky: 1.99 to 3.46 W m<sup>-2</sup>), these contrasts indicate that the sign of the net anomaly is controlled by the relative magnitudes of the SW and LW anomalies. The negative net anomaly over USW is driven by a comparatively large increase in outgoing LW radiation that outweighs the reduced outgoing SW, a situation that can arise from enhanced surface or lower-tropospheric emission or from reductions in cloud amount that raise outgoing LW. The positive net anomaly over NA is driven primarily by reduced outgoing SW combined with only modest LW changes, yielding an increase in downward energy at the TOA. The uniformly positive net anomalies in the transport regions reflect cases where reductions in outgoing SW and reductions or only small increases in outgoing LW act together to increase the downward radiative fluxes. Furthermore, in some transport sectors aerosol-cloud interactions that alter cloud optical properties and cloud-top temperature further modulate the LW contribution and reinforce the positive net.

535

540

Figure 10. Wildfires are associated with Clear-sky TOA radiative Flux Changes. (a-i) Monthly climatological mean TOA radiative fluxes (a, d, g), mean TOA radiative fluxes during Wildfire events (b, e, h) and mean wildfire-associated TOA heat flux anomalies (c, f, i) for SW radiation (a-c), LW radiation (d-f) and Net radiation (g-i). All radiative fluxes are defined as positive upward at the TOA: a positive SW anomaly indicates increased outgoing solar radiation or reduced cloud reflection; a positive LW anomaly means more outgoing LW radiation is retained less emitted to space. Net is qualitatively by using SW and LW, but net is calculated independently to remove the seasonal effects of wildfire frequency. Anomalies represent the difference between total events and climatological mean events (Wildfires - Climatology), calculated independently to remove seasonal variations in wildfire events (see in Section 2.3.2).

Table 2 and 3 present the regional averages of TOA SW, LW, and net radiative fluxes anomalies during the wildfire period, for both all-sky and clear-sky conditions. The regions are divided into the wildfire-origin regions (WOR: USW and NA) and long-range transport regions (LTR: NSCAN, WE, and IGB). Our computed of SW radiative Flux Change -4.48  $\pm$  0.08 W m<sup>-2</sup> in TOA radiation over NA region by smoke aerosols is close to the estimate of -4.40  $\pm$  2.5 W m<sup>-2</sup> reported by Pfister et al. (2008) using CERES datasets with difference in TOA Clear-sky Net, SW and LW fluxes between the summers 2004 and 2000 in the Alaska (Table 2). Within such change, clear-sky of -3.78  $\pm$  0.22 W m<sup>-2</sup> and all-sky of 0.47  $\pm$  0.28 W m<sup>-2</sup> in TOA radiation over USW region is close to the estimate of -3.33  $\pm$  0.89 W m<sup>-2</sup> and 1.35  $\pm$  1.80 W m<sup>-2</sup> by Thornhill et al. (2018) using two simulations of high and low emissions of BBA. Previous studies have reported substantial radiative impacts of the 2019/20 Australian wildfires. For example, Khaykin et al. (2020) estimated an equinox-equivalent regional radiative forcing of approximately -1.0 W m<sup>-2</sup> at the TOA, with a global cloud-free mean of -0.31  $\pm$  0.09 W m<sup>-2</sup>, indicating a net cooling effect driven by stratospheric smoke aerosols.In contrast, our analysis of the 2021 North American wildfires reveals regionally opposite TOA net anomalies. Under clear-sky conditions, USW exhibits a -1.70  $\pm$  0.49 W m<sup>-2</sup> anomaly, whereas NA shows a

 $+2.65 \pm 0.24$  W m<sup>-2</sup> anomaly. When clouds are included (all-sky), these magnitudes are amplified to  $-2.50 \pm 0.25$  W m<sup>-2</sup> (USW) and  $+5.62 \pm 0.23$  W m<sup>-2</sup> (NA).

These contrasting signs suggest that cloud-aerosol interactions and differences in aerosol absorption dominate the regional radiative responses, with absorbing smoke layers above clouds likely enhancing TOA warming in northern regions, while more scattering-dominated plumes over the US West Coast induce cooling.

Figure 11. Wildfires are associated with All-sky TOA radiative Flux Changes. (a-i) Monthly climatological mean TOA radiative fluxes (a, d, g), mean TOA radiative fluxes during wildfire events (b, e, h) and mean wildfire-associated TOA heat flux anomalies (c, f, i) for SW radiation (a-c), LW radiation (d-f) and Net radiation (g-i). All radiative fluxes are defined as positive upward at the TOA: a positive SW anomaly indicates increased outgoing solar radiation or reduced cloud reflection; a positive LW anomaly means more outgoing LW radiation is retained less emitted to space. Net is qualitatively by using SW and LW, but net is calculated independently to remove the seasonal effects of wildfire frequency. Anomalies represent the difference between total events and climatological mean events (Wildfires - Climatology), calculated independently to remove seasonal variations in wildfire events (see in Section 2.3.2).

Table 2. Regional Mean Clear-Sky TOA SW, LW, and Net Radiative Fluxes during Wildfires compared with Previous Studies (W m-2).

Khaykin et al. (2020) (2019/20 Australian  $-0.31\pm0.09$ Australian WOR Thornhill et al. (2018) South American  $-3.33\pm0.89$  $0.53\pm0.93$ (High-Low experiments) BBR Pfister et al. (2008) (2004 - 2000)-2.5±5.6 -4.4±2.5 -6.9±6.2 Alaska SW\_up anomaly -3.78 $\pm$ 0.22 -4.48 $\pm$ 0.08 -2.21 $\pm$ 0.27 -7.07 $\pm$ 0.22 -3.99 $\pm$ 0.31  $3.47\pm0.81$   $1.53\pm0.34$   $0.01\pm0.38$  $3.57\pm0.44$ IGB (2024—Climatological Means (2005—2015))  $-1.70\pm0.49$   $2.65\pm0.24$   $-0.18\pm0.99$   $4.57\pm0.42$ WE This study NSCAN LW\_up anomaly 4.95±0.42 1.44±0.24 NA WOR MSDRadiation type Ward anomaly Net down-TOA

Table 3. Regional Mean All-Sky TOA SW, LW, and Net Radiative Fluxes during Wildfires compared with Previous Studies (W m<sup>-2</sup>).

|                           | (20        | v24—Climato | This study (2024—Climatological Means (2005—2015)) | ; (2005–201; | 5))        | Thornhill et al. (2018) Jouan et al. (2024) Khaykin et al. (2020)  (High—Low (2002 - 2021 (2019/20 Australian | Jouan et al. (2024)<br>(2002 - 2021 | Khaykin et al. (2020)<br>(2019/20 Australian |
|---------------------------|------------|-------------|----------------------------------------------------|--------------|------------|---------------------------------------------------------------------------------------------------------------|-------------------------------------|----------------------------------------------|
| Radiation type            | M          | WOR         |                                                    | LTR          |            | experiments) BBR                                                                                              | period)<br>LTR                      | Wildines) WOR                                |
|                           | MSM        | NA          | NSCAN                                              | WE           | IGB        | South American                                                                                                | $SEA^a$                             | Australian                                   |
| TOA                       |            |             |                                                    |              |            |                                                                                                               |                                     |                                              |
| SW_up<br>anomaly          | 0.47±0.28  | 3.58±0.28   | 3.58±0.28 -3.54±0.29 3.12±0.36 0.35±0.38           | 3.12±0.36    | 0.35±0.38  | 1.35±1.80                                                                                                     | +7.22 ± 4.26                        | I                                            |
| LW_up<br>anomaly          | 3.30±0.22  | -8.15±0.23  | -8.15±0.23 2.62±0.14 -5.23±0.18 -2.66±0.24         | -5.23±0.18   | -2.66±0.24 | 3.07±1.55                                                                                                     | I                                   | I                                            |
| Net down-<br>Ward anomaly | -2.50±0.25 | 5.62±0.23   | 5.62±0.23 1.99±0.30 3.33±0.34 3.46±0.44            | 3.33±0.34    | 3.46±0.44  | I                                                                                                             | I                                   | -1.0                                         |

<sup>a</sup> Induced by biomass burning aerosol (BBA) transported from southern Africa to the south-eastern Atlantic (SEA) stratocumulus region.

#### 4.3 Influence of cloud radiative effect on TOA radiation

Figure 12 shows anomalies in TOA radiative fluxes and CRE that coincide with observed smoke plumes; these anomalies are consistent with smoke-induced modifications to cloud properties. The TOA SW CRE, representing the ingoing SW fluxes perturbation, exhibits strong negative anomalies over most smoke-affected regions, consistent with enhanced SW reflection due to aerosol-cloud interactions. In particular, USW shows an anomalously weak SW CRE near 0 W m<sup>-2</sup>, significantly lower than surrounding regions. This suggests reduced cloud-mediated cooling likely due to partial cloud dissipation or smoke-driven semi-direct effects suppressing cloud formation. In contrast, other fire-origin regions such as the high-latitude regions of NA show SW CRE anomalies of -30 to -40 W m<sup>-2</sup>, while WE also exhibits negative SW CRE values of -20 to -35 W m<sup>-2</sup>, indicating substantial enhancement in SW reflection from wildfire-influenced cloud systems.

Figure 12. TOA CRE Changes associated with wildfires: Mean clear-sky (a) and all-sky (b) TOA outgoing SW Flux, mean TOA outgoing SW CRE (c), mean clear-sky (d) and all-sky (e) TOA outgoing LW Flux, mean TOA outgoing LW CRE (f), mean clear-sky (g) and all-sky (h) TOA net flux during wildfire events, and mean TOA net CRE (i). CRE (SW, LW, net) was calculated as the difference between all-sky and clear-sky fluxes, weighted by cloud fraction.

The TOA LW CRE displays an opposite but spatially coherent pattern. In the USW, LW CRE reaches +45 to +50 W m<sup>-2</sup>, indicating strong LW trapping due to elevated cloud tops and optically thick cloud layers induced by pyro-convective activity. Similarly, NA shows localized LW CRE enhancements, while England, Northern Scandinavia, and parts of Western Europe exhibit widespread LW CRE values of +30 to +50 W m<sup>-2</sup>, reflecting the regional-scale impact of smoke-laden,

vertically developed cloud systems on outgoing LW radiation. When integrated across all-sky and clear-sky fluxes, the regional mean TOA net flux reaches  $+63.59 \pm 0.52$  W m<sup>-2</sup>, while the clear-sky baseline is  $+36.72 \pm 0.55$  W m<sup>-2</sup>, yielding a mean net CRE of  $-26.87 \pm 0.75$  W m<sup>-2</sup>. This reflects a net cooling effect over the entire analysis domain driven predominantly by cloud-mediated SW enhancement. Notably, the USW exhibits localized positive net CRE, indicating a rare case of net warming, likely due to weak SW reflection but strong LW trapping. In contrast, the England and Northern Scandinavia regions display pronounced negative net CRE, consistent with enhanced albedo and reduced outgoing energy fluxes.

Figure 13 illustrates the daily evolution of TOA CRE across five study regions in August 2024: USW, NA, NSCAN, WE, and IGB. These data, derived from CERES retrievals, represent the net radiative impact of cloud cover for SW, LW, and net radiation components (Figure 13-14). During the early phase of widespread wildfire activity (August 6-11, red-shaded area), the two primary fire source regions (the US West Coast and Northern NA) exhibit relatively stronger net CRE values compared to the downwind regions. This is indicative of enhanced cloud-top warming effects and reduced net cooling, possibly driven by the injection of absorbing aerosols and their semi-direct effects, or a temporary suppression of COD.

During the PyroCb development and lofting phase (August 11-14, blue-shaded), NA and USW shows a marked peak in net CRE, reaching a maximum of +8.48 W m<sup>-2</sup>, indicating a transient but substantial warming effect likely linked to deep convection, ice-phase cloud development, and altered cloud vertical structure associated with PyroCb outbreaks. Meanwhile, the downwind regions (NSCAN, and IGB) maintain more negative CRE values (-40 to -100 W m<sup>-2</sup>), consistent with typical marine stratiform cloud regimes uninfluenced by immediate fire activity.

In the later stage of the month (August 14-22, gray-shaded), corresponding to the long-range transport of wildfire aerosols ("smoke aerosol conveyor"), CRE in source regions gradually returns toward baseline, while downwind regions continue to experience persistent radiative cooling. This lagged response suggests the prolonged influence of aged smoke aerosols on cloud radiative properties and emphasizes the importance of interregional transport pathways in modulating TOA energy budgets. Figures 12 and 13 demonstrate the spatial footprint and temporal evolution of wildfire-driven perturbations to cloud radiative forcing. They highlight the dual-phase response: initial CRE enhancement near fire sources followed by extended radiative cooling in remote receptor regions, providing compelling observational evidence for the far-reaching climatic influence of extreme fire events. This sustained negative CRE changes signal aligns over NA with the enhanced cloud reflectivity and reduced ΔCER documented in Figure 7, supporting the presence of smoke-induced microphysical effects such as the Twomey effect.

625

Figure 13. Time-series of CERES cloud radiative effect changes in the 5 study regions (blue and gray rectangles in Figure 1) in August 2024. Red shading marks wildfire activity; blue shading marks PyroCb development; gray shading marks smoke aerosol conveyor activity.

## 4.4 PyroCb-Driven Transcontinental Transport and Cloud-Radiation Effects

In this study, we investigate the spatiotemporal evolution of wildfire activity based on satellite-derived active fire masks and MaxFRP, offering a high-temporal-resolution perspective on daily fire intensity. We examine the meteorological environment and large-scale circulation that modulate the emission and dispersion of fire-related aerosols, with particular attention to the vertical transport processes that facilitate long-range propagation of smoke plumes. Furthermore, we assess the influence of smoke aerosols on cloud formation and vertical structure using synergistic observations from passive and active satellite sensors. Finally, we quantify the TOA radiative impacts of fire-induced aerosol-cloud interactions, employing physical diagnostics such as CRE and net radiative flux perturbations. By integrating multi-platform satellite observations with reanalysis data, our study offers new insights into the atmospheric and radiative consequences of extreme biomass burning events and provides a comprehensive methodological framework for coupled fire-aerosol-cloud-radiation analysis (Figure 14).

During the North American wildfires in August 2024, pronounced anomalies in radiative fluxes were observed over both the fire source and the downwind smoke-affected regions (Figure 14). SW radiation exhibited positive anomalies of +1.94 W m<sup>-2</sup> over the fire-source region and +0.53 W m<sup>-2</sup> over the downwind smoke-affected region, consistent with enhanced scattering and reflection of solar radiation by smoke aerosols and aerosol-cloud modifications. In contrast, outgoing LW radiation decreased by -3.62 W m<sup>-2</sup> over the fire-source region and -2.48 W m<sup>-2</sup> over the smoke-affected area, reflecting changes in

infrared emission associated with wildfire-perturbed cloud and plume structure. Overall, these net anomalies correspond to a reduction in outgoing radiation, they are equivalent to downward radiative forcings of +1.68 W m<sup>-2</sup> and +1.95 W m<sup>-2</sup>, respectively, indicating that the LW response dominates and produces a net warming effect on the local column. Corresponding increases in ΔCOD and ΔCTH increased over both regions (+3.39 and +3.89 for ΔCOD; +3.2 and +3.7 for ΔCTH in fire-source and smoke-affected areas, respectively), reflecting the presence of low-temperature clouds. This aligns with the observed suppression of outgoing LW radiation.

The radiative effects of smoke aerosols exhibited contrasting behaviours across the fire-source regions. Over the USW area, TOA upward fluxes increased by +0.47 W m<sup>-2</sup> (SW) and +3.30 W m<sup>-2</sup> (LW), indicating a net increase in outgoing radiation and a modest cooling effect. In contrast, the NA region showed a strong reduction in outgoing LW radiation -8.15 W m<sup>-2</sup> accompanied by an increase in outgoing SW flux +3.58 W m<sup>-2</sup>, indicative of pronounced infrared trapping by high, cold clouds or thick smoke layers. Consequently, the reported net downward anomaly was negative -2.50 W m<sup>-2</sup> in USW but positive +5.62 W m<sup>-2</sup> in NA, reflecting opposite radiative regimes dominated by LW cooling in USW and LW warming in NA.

From the perspective of cloud microphysical parameters, the ΔCER in the fire source region decreased significantly by 3.11 μm, indicating that aerosols suppressed cloud droplet growth by increasing the number of cloud condensation nuclei—a classic "first indirect effect." ΔCOD markedly increased, enhancing cloud reflection of SW radiation and further intensifying cooling. ΔCTH rose, indicating that fire plumes and intense convection promoted the formation of deep clouds, facilitating aerosol transport to upper levels. Concurrently, the CRE was negative, meaning clouds overall cooled the energy balance by enhancing SW reflection and weakening LW greenhouse effects. The combined effects of ΔCER, ΔCTH, and ΔCOD create a stark contrast between cloud cooling and radiative heating, highlighting the crucial regulatory role of aerosol-cloud-radiation interactions during wildfire events.

Figure 14. Schematic diagram of PyroCb-driven transcontinental transport of North American wildfire aerosols and their aerosol-cloud-radiative effect.

## **5 Conclusions**

This study provides a comprehensive satellite-based analysis of wildfire-induced atmospheric disturbances during the August 2024 North American wildfire outbreak, with a particular focus on wildfire induced aerosol-cloud-radiation effects. By integrating multi-source satellite datasets (MODIS/MOD14, Sentinel-5p/TROPOMI, GOES-16/ABI, Suomi-NPP/OMPS, Suomi-NPP/VIIRS, EarthCARE/CPR, PACE/SPEXOne and CERES) with MERRA-2 reanalysis and NOAA forward trajectory modelling, we characterized the evolution of fire activity, aerosol vertical transport, cloud microphysical perturbations, and TOA radiative impacts.

Our results indicate that intense wildfire activity (quantified using MODIS fire pixels counts and MaxFRP) is significantly correlated with elevated CO concentrations and deep vertical motion. Combined with ultraviolet aerosol index (UVAI) and EarthCARE CPR radar observations, these results suggest that wildfire convection can inject heat into the upper troposphere. CO anomalies originating from TROPOMI combined with vertical velocity diagnostics confirm the effective suspension and transcontinental transport of the biomass burning plume, especially during the PyroCb event. In the downwind direction, the

https://doi.org/10.5194/egusphere-2025-5076 Preprint. Discussion started: 14 November 2025

© Author(s) 2025. CC BY 4.0 License.

cloud response is evident from the CERES-derived ΔCTH, ΔCOD, and ΔCER anomalies, which are consistent with aerosolcloud interactions and the Twomey effect. Notably, CRE and net radiative forcing show significant spatial variability:
regions such as Western Europe and northern Scandinavia experience enhanced SW radiation and LW trapping, altering the
TOA energy budget. Analysis of ΔNet anomalies relative to climatology further emphasizes the persistent radiative
perturbations caused by wildfire aerosol transport.

These findings have several important implications. First, they reveal that intense wildfires can drive vertical aerosol injection and alter cloud microphysical properties far beyond the fire source area. Second, clouds over these fire regions simultaneously produce both negative and positive radiative effects—enhancing SW reflection while either weakening LW greenhouse effects or amplifying LW radiation effects. Furthermore, these smoke aerosols induce comparable radiative effects, with warming reaching +2.84 W m<sup>-2</sup> over the fire source region and a more pronounced warming impact of +3.16 W m<sup>-2</sup> in distant transport zones. The spatial distribution pattern of the warming radiative anomaly highlights the radiative significance of smoke-cloud interactions near fire sources and in downwind environments. This has critical implications for regional climate feedbacks, atmospheric heating rates, and potential precipitation regulation.

However, several limitations should be acknowledged. The spatial and temporal coverage of polar-orbiting sensors constrains continuous monitoring of rapidly evolving convective events. Uncertainties also arise from the differing retrieval algorithms and spatial resolutions of the satellite products employed. Moreover, the radiative effects inferred from satellite observations are observationally constrained and do not fully capture underlying dynamical feedbacks without coupling to models. Future research should pursue in the directions of integrate high-resolution numerical simulations with satellite observations to quantify feedback loops between aerosols, clouds, and radiation.

https://doi.org/10.5194/egusphere-2025-5076 Preprint. Discussion started: 14 November 2025

© Author(s) 2025. CC BY 4.0 License.

EGUsphere Preprint repository

Data availability: CERES SYN1deg data used in this analysis are available at https://ceres.larc.nasa.gov/. MODIS MOD14A1 and MCD64A1 products are available from https://modis.gsfc.nasa.gov/data/. MERRA-2 reanalysis data can be accessed at https://gmao.gsfc.nasa.gov/reanalysis/MERRA-2/. GOES-16 OR\_ABI-L2-MCMIPF-M6 data are available from https://noaa-goes16.s3.amazonaws.com/. Suomi-NPP/VIIRS L1B 6-min swath imagery is available at https://ncc.nesdis.noaa.gov/. OMPS/NPP NMMIEAI-L2 products are available at https://omps.gsfc.nasa.gov/. Sentinel-5P/TROPOMI L2 CO data can be accessed via https://s5phub.copernicus.eu/. PACE SPEXone L2 RemoTAP data are available at https://public.spider.surfsara.nl/project/spexone/. ECA\_JXCA\_CPR\_NOM\_1B data are available at https://earthcarehandbook.earth.esa.int/catalogue/ac\_\_tc\_\_2b/. For any additional datasets or processed data not publicly available, requests can be directed to the corresponding authors.

Author contribution: Y. W. and C. C. proposed the initial idea for this paper. Y. J. C. retrieved aerosol-related data. H. X. Y. provided insights on aerosol-cloud interactions. Y. Y. C., Q. Y. H., S. W., Z. Q. L. and C. Z. provided insights about the analysis of aerosol long-range transport. Y. W. and C. C. wrote the manuscript based on comments from all authors.

Ackowledgement: This research was supported by the National Key R&D Program of China (grant no. 2024YF0811200). The authors gratefully acknowledge the various data providers that made this study possible. The CERES SYN1deg and MERRA-2 datasets were produced by NASA's Earth Science Division and distributed through the NASA Langley Research Center and the Global Modeling and Assimilation Office (GMAO), respectively. MODIS active fire (MOD14A1) and burned area (MCD64A1) products were provided by the NASA MODIS Science Team. GOES-16 ABI Level-2 data were obtained from NOAA/NESDIS, and Suomi-NPP/VIIRS L1B imagery was provided through the NOAA CLASS archive. We also acknowledge NASA Goddard Space Flight Center for providing OMPS/NPP and PACE/SPEXone (RemoTAP) data, and the European Space Agency's Copernicus program for access to Sentinel-5P/TROPOMI Level-2 products. The authors further thank the EUMETSAT and JAXA EarthCARE mission teams for the CPR Level-1B (ECA\_JXCA\_CPR\_NOM\_1B) dataset.

**Competing interests:** The authors declare that they have no conflict of interest.

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
