# Peer review of "Wildfire aerosols lofted by North American pyrocumulonimbus clouds: long-range transport and aerosol-cloud-radiative effects"

_EGUsphere, 2025_

## Referee Comment (RC2)

This study investigates the long-range transport of wildfire aerosols and their effects on cloud and radiation during an intense North American wildfire event in August 2024. Multiple sources of observations from satellites and reanalysis products have been used, especially the recently launched PACE and EarthCare. They end up with a quantification of radiation effect of fire aerosols.

The topic is of importance and it's an interesting study with a lot of work clearly done; but I believe its structure could be further improved and some sections would benefit from additional clarification. The methodology, especially the approach for isolating fire and non-fire influences (see detailed comments below), also requires refinement.

Overall I'd be positive on the manuscript if the following concerns can be satisfactorily addressed.

**Main comments:**

Introduction: To improve the manuscript's flow, the authors could include a brief preview of each section's content.

I feel the structure of the manuscript can be improved. Section 4 should be included in the result section (#3).

L278: strong vertical uplift: the positive value of vertical velocity means downward motion not uplift, which seems to contradict the conclusion drawn from the data

Fig. 2: how did the authors extract the fire-induced changes in AOD and BC from background?

L334-336: Earlier the authors stated the 4-11 um difference can identify the smoke aerosols above convective cloud-top, implying the direct effect, while here they instead attribute to the changes in cloud microphysics. They appear to be contradictory. How did the authors separate the contributions from direct effect of smoke aerosols and and indirect effect via change of clouds?

Fig. 4 shows that the smoke plume needs around 1 week to travel to Europe. It would be interesting to check if there is an obvious increase in AOD at the destinations of air parcels. This figure seems more suitable for section 3.2 than section 3.1

**Fig.7**: The authors attributed the changes in cloud properties relative to multi-year climatology to the fire-induced aerosols – I'm not sure if it's true. Clouds, particularly the convective clouds, are modulated primarily by dynamic background, then affected by aerosols via microphysical processes. The difference in Fig.7 reflects combined effects of meteorological conditions (as shown in Fig. 5) and aerosols, not aerosols alone. A good example is the similar patterns of \Delta CTH and wind field (Fig.5) on 17 Aug. Although the meteorological background on 15 Aug wasn't shown, I'd assume the strong positive anomaly at (25N, 70W) corresponds to a cyclone system, which is nothing to do with the fire-induced aerosols. Also, some suggestions/comments to this figure:

    - It's not easy to follow the text with the current form of figure, as the locations of 'source region' and 'intense PyroCb development' are not obvious to readers. I suggest to mark the locations affected by fire aerosols – one can make use of the information from geostationary satellites. I assume most descriptions in the text, e.g., 'ΔCTH exhibited marked increases of up to ~5 km', are meant for the mean value of the entire region, right? Would be good to report values only in the fire-relevant areas. Importantly, by look at the target areas, one can, at least, eliminate the confounding effect from meteorological background in non-fire regions. This suggestion also applies to Figure 10,11,12.

    - PyroCb is a deep convective system that should have a cold/ice top, where CER is not relevant and cannot even be successfully retrieved from satellite. Please clarify.

- What is the motivation to select 13,15,17,19 Aug, especially when 10 Aug had the strongest fire event?

**Minor comments:**

L21-23: It's unclear if those changes appear in the PyroCbs or the downwind clouds. Please clarify.

L52: Twomey effect → the Twomey effect

L63: fires and emissions: what 'emissions' do you mean?

L64-65: the formulation is a bit confusing. The year 2024 appears to be the highest emission year, but the text seems to point to 2023..

L69: demonstrating that these fires → which, or split this sentence into two.

L71: criteria is not the right word here. I'd suggest 'providing evidence for' or simply 'supporting'

Fig.1: what does 'cloud albedo effect' (the unit is missing here) mean? Cloud albedo effect literally means the Twomey effect,; however, I don't think the Twomey effect is provided by CERES_SYN1deg-Month product. Also, how was the cloud albedo effect used to determine the study regions?

L108-109: Each instrument should come with a reference.

Table 1: Please clarify which cloud parameters are from PACE/SPEXone and what 'data' are from GOES-16/ABI and suomi (I assume it's reflectence). Also, SPEXone is designed for aerosol retrievals. I don't think it officially provides cloud properties. Harp2 or OCI would be the right instrument to look at.

L127: full name of BT/BTD

L129: what is Climate Radiative Effect (CRE)? It's not a common term to use. But looking at Eq. 5, it seems Cloud Radiative Effect but with an additional term F_cloud. Please clarify what the CRE exactly is, preferably provide relevant references where this definition comes from.

Eq. 1: is there a reference supporting this equation? Please explain the meaning of 'A' and 'N'.

L148-149: please elaborate a bit on how the fire intensity was derived and what criteria were used for fire hotspots classification.

L164-169: PM2.5 is mass concentration (ug m-3), not 'aerosol loading'. Loading commonly means burden. Also, kg kg-1 corresponds to mass mixing ratio, not concentration. I think you need to convert kg kg-1 for each aerosol species to ug m-3 before applying Eq. 2.

L255: framework → equation

Eq. 7: I didn't get the point of using spatially aggregated radiative forcing. We always use mean forcing to assess the climate effect not the aggregate one. Additionally, it appears that the term (Netpert - Netmean) is equivalent to ΔNet in Equation 6. If this is the case, please reorganize the two equations for clarity.

Lines 258-259: This metric is just a forcing term. The claim that it can constrain climate sensitivity requires further justification.

L265-266 vs L64-65: I'm even more confused which year has the highest CO2 emission. Please clarify.

Fig.2: which vertical level was selected for RH and vertical velocity?

The captions of Fig.2 (c) & (d) are incomplete. Without diving into the text, one wouldn't know what are being shown, particular if two figures are the same but only date/period differs. I also feel the figure titles are a bit misleading. Furthermore, the definition of 'anomaly' (particularly the reference period) is missing. Also, firepots are missing in high AOD area, undermining the argument in L296-297.

L315-316: The aerosol, cloud, and precipitation components should be clearly labeled or described here. I didn't really see a convective system; it looks like the black band near the boundery layer is cloud part and above is aerosol. The cloud structure is not obvious in true color images in Fig.3 a & b, as UV AI masks most areas. Overlaying AI contour lines might help the visualization.

L339-L341: The claimed 'increase in albedo' in Figure 4 is not visually apparent. A quantitative metric would help.

Fig. 5: Please explain what the red line means in the caption and mark the fire regions in the plot.

Fig. 6: Table 1 shows CO is from TROPOMI while here authors stated it's from MERRA2. Please correct the wrong one.

L386: latitudinal → longitudinal

Fig.9: Are the date from SPEXone as well?

L458-469: Using the simple temporal evolution of regional means to explain cloud fraction responses is too simplistic. Especially, the 5-day delay of response of CF is unusual. I suggest to omit the discussion on cloud response if there is no strong evidence. Similarly for L471-475, the Twomey effect is also not justified here without showing CDNC results.

---

## Community Comment (CC1)

Wang et al. (W25) have offered an intriguing analysis of pyroCb effects on long-range transport and downstream effects. Reading the manuscript with great interest, I encountered a puzzling situation worth bringing to the authors' attention.

Their Figure 4 is the focal point for a detailed illustration of pyroCb-generated clouds and trajectories launched therefrom, in order to make a long-range connection with their Europe focal zones. Whereas W25 appropriately invoke (and cite) an established pyroCb-cloud-detection method combining window IR brightness temperature (BT) and shortwave-window IR brightness temperature difference (BTD), their illustration in Figure 4 comes up short in multiple respects. This came to my attention because the routine, pyroCb monitoring performed within the Worldwide PyroCb Information Exchange (https://groups.io/g/pyrocb) resulted in no pyroCb events in the northwestern USA or southwestern Canada on 12 August 2024 (local time). See Peterson et al. (2025) for a description of the WPIE methodology. Two pyroCbs were identified, but in Northwest Territories north of 60◦N.

I reviewed GOES data for evidence of pyroCbs closer to Figure 4's target features and thereby discovered that the GOES imagery shown in Figure 4 does not apply to the date/time given in the labeling and caption (00:10 and 00:50 UTC 13 August 2024). An example of GOES 16 Channel 13 BT at 00:10 UTC 13 August 2024 (Day number 226) is shown below.

[Figure]

Comparing this with Figure 4a-b reveals a totally different cloud landscape between the two. Thus it appears that the Figure 4 imagery is in error. I checked imagery from 24 hours earlier and later and did not find anything resembling a match with Figure 4. That raises the question of when the cloud fields in Figure 4 occurred. To the extent that the Figure 4 image and trajectory analysis informs W25's findings and conclusions, it appears that this apparent error should be addressed.

Another perceived shortcoming of Figure 4 and attendant discussion is that no fires were identified as the source of pyroCb convection. Given the fact that the WPIE did not document any pyroCbs near the launch points of Figure 4's trajectories, it is suggested that any new source-receptor analysis performed by W25 include a pyroCb image analysis that includes identification of the fire generating the pyroconvection. Lastly, it would be helpful for W25 to use trajectory launch altitudes representative of the cloud heights inferred from the GOES BT imagery. Presently, W25 use 3, 5, and 7 km above ground level, which are low relative to expectations for exhaust from a mature pyroCb.

**References**

Peterson, D.A., Berman, M.T., Fromm, M.D. *et al*. Worldwide inventory reveals the frequency and variability of pyrocumulonimbus and stratospheric smoke plumes during 2013–2023. *npj Clim Atmos Sci* **8**, 325 (2025). https://doi.org/10.1038/s41612-025-01188-5

---

## Community Comment (CC2)

**We appreciate the community commenter (Dr. Michael Fromm) thoughtful and scientifically relevant comments, which helped improve the clarity and robustness of the manuscript. Our detailed responses are given below.**

Community Commenter: Wang et al. (W25) have offered an intriguing analysis of pyroCb effects on long-range transport and downstream effects. Reading the manuscript with great interest, I encountered a puzzling situation worth bringing to the authors' attention.

**Response: We thank the Community Commenter for their careful reading of the manuscript and for bringing this issue to our attention. Below, we clarify the source of this apparent discrepancy and describe how it has been addressed in the revised analysis.**

Their Figure 4 is the focal point for a detailed illustration of pyroCb-generated clouds and trajectories launched therefrom, in order to make a long-range connection with their Europe focal zones. Whereas W25 appropriately invoke (and cite) an established pyroCb-cloud-detection method combining window IR brightness temperature (BT) and shortwave-window IR brightness temperature difference (BTD), their illustration in Figure 4 comes up short in multiple respects. This came to my attention because the routine, pyroCb monitoring performed within the Worldwide PyroCb Information Exchange (https://groups.io/g/pyrocb) resulted in no pyroCb events in the northwestern USA or southwestern Canada on 12 August 2024 (local time). See Peterson et al. (2025) for a description of the WPIE methodology. Two pyroCbs were identified, but in Northwest Territories north of 60°N.

**Response: We appreciate the reviewer's reference to the Worldwide PyroCb Information Exchange (WPIE) monitoring results. Following the reviewer's comment, we re-evaluated the timing and location of the pyroCb activity shown in Figure 4 using the corrected ABI geolocation and an expanded set of independent constraints.**

**Using the revised analysis, we confirm that no pyroCb activity is identified over the northwestern United States or southwestern Canada on 12 August 2024 (local time), consistent with the WPIE records. The pyroCb features discussed in the manuscript are confined at 23:50 UTC, August 13 2024, with the primary source region located over northern Alberta (Region 1 at 59.5°N, 110.4°W). The timing and location of this event are consistent with the pyroCb activity formed at 00:00 UTC, August 14 on a fire at 59.4°N, 110.2°W and under the thin anvil of a regular Cb reported by WPIE.**

**The revised manuscript text addressing this point is provided below for clarity.**

**[Revised manuscript text:]**

"

[Figure]

**Figure 4. (Figure 1. in this paper) ABI/GOES-16 observations of cloud-top evolution and smoke transport associated with the 13 August 2024 North American pyroCb events. (a-b) Brightness temperature at 11 μm (BT11) and (c-d) brightness temperature difference between 4 and 11 μm (BTD4-11) derived from GOES-16/ABI on 13 August 2024. Red dots denote MOD14A1 active fire**

detections, and yellow shaded areas indicate the identified intense pyrocumulonimbus (IPCB) source regions. The pink circle represents the 50 km radius used for pyroCb statistical analysis, with the cyan dot marking the circle center. (e-f) NOAA HYSPLIT Forward trajectories of air parcels released from the pyroCb source regions on 13-14 August 2024 and transported to the downwind smoke conveyor region. (a,c,e) Region 1 at 59.6°N, 110.8°W and Region 2 at 56.4°N, 104.7°W, with trajectories initialized at 23:00 UTC on August 13 and arriving at the target region by 23:00 UTC on August 19. (b,d,f) Region 1 at 59.5°N, 110.4°W , Region 2 at 56.8°N, 104.6°W and Region 3 at 59.5°N, 120.95°W, with trajectories initialized at 00:00 UTC on August 14 and arriving at the target region by 00:00 UTC on August 20.

Figure 4a-d shows GOES-16/ABI observations on 13 August, revealing two distinct regions of fire-associated deep convection over northern Alberta and northern Saskatchewan, Canada. The BT11 imagery (Figure 4a-b) highlights exceptionally cold cloud tops, with darker shading corresponding to lower temperatures and greater cloud-top altitudes, consistent with intense pyroCb convection. These features are colocated with pronounced positive BTD4-11 enhancements (Figure 4c-d), reflecting altered cloud-top radiative properties associated with smoke aerosol entrainment and reduced effective particle sizes.

The thermally and microphysically distinct cloud tops are spatially coincident with active fire detections, providing strong observational evidence for a direct linkage between surface fire activity and the observed deep convective development. Yellow shaded areas in Figure 4c-d indicate IPCB source regions that satisfy all identification criteria and represent robust detections of pyroCb-injected air masses (Li et al., 2025). Following convective lofting, these air masses undergo sustained upper-tropospheric transport and gradual eastward advection under synoptic-scale flow, illustrating the combined roles of intense fire-driven convection and large-scale circulation in enabling long-range smoke transport across central and eastern Canada.

Based on three GOES-16 infrared channel observations in Figure 4a-d, pyroCb were identified as convective clouds anchored to the wildfire source, the specific locations of pyroCb formation were identified as the yellow regions within the overlapping areas between the black regions in the grayscale shaded map and the blue-green highlighted regions. Figure 4e-f presents the NOAA HYSPLIT forward trajectories calculated to investigate the long-range transport of air masses originating from intense IPCB regions. The trajectories reveal rapid vertical lofting associated with intense pyroconvection, followed by sustained advection in the upper troposphere and lower stratosphere (UTLS). The initial release heights were constrained using cloud vertical structures

**from CPR/EarthCARE observations and cloud top height information from the ATLID lidar.**

**At 23:10 UTC on 13 August, two IPCB source regions were identified. Forward trajectories were initialized at 4, 8, and 11 km above ground level (AGL). Air parcels released at 4 km AGL experienced rapid vertical uplift to 6-8 km, consistent with strong convective injection associated with IPCB activity, whereas air masses initialized at 8 and 11 km AGL largely maintained transport altitudes above 9 km, indicating efficient entrainment into the UTLS and subsequent isentropic transport. At 23:50 UTC on 13 August, trajectories were initialized from the two established IPCB source regions and an additional developing source region at 4, 7.5, and 10 km AGL. Pronounced vertical lifting from 4 km to 6-8 km remained confined to the established IPCB regions, while air parcels released at higher altitudes were transported predominantly at around 7 km, indicating sustained mid-to-upper tropospheric transport. Vertical uplift associated with the developing source region was comparatively limited. By 06:00 UTC on 14 August, forward trajectories from multiple IPCB regions showed localized convergence and partial merging, implying coordinated airflow and plume aggregation. Following this merging phase, the combined air masses were advected eastward large-scale synoptic circulation, crossing the North Atlantic and reaching northern and western Europe, characteristic of intercontinental UTLS transport of air masses lofted by deep pyroconvection."**

I reviewed GOES data for evidence of pyroCbs closer to Figure 4's target features and thereby discovered that the GOES imagery shown in Figure 4 does not apply to the date/time given in the labeling and caption (00:10 and 00:50 UTC 13 August 2024). An example of GOES 16 Channel 13 BT at 00:10 UTC 13 August 2024 (Day number 226) is shown below.

[Figure]

**Response: We thank the reviewer for this careful and constructive comment. We agree that the discrepancy identified between Figure 4 and independent ABI imagery indicates a problem in the original version of Figure 4, and we appreciate the reviewer's effort in cross-checking imagery from adjacent time periods.**

**Upon re-examination of our data processing workflow, we determined that the issue originated from an incomplete numerical implementation of the GOES-16 ABI fixed-grid projection to latitude–longitude transformation used to generate Figure 4. Although the reprojection was based on the standard formulation described in the "GOES-R SERIES PRODUCT DEFINITION AND USERS' GUIDE (PUG, Volume 5, Section 4.2.8)" and the reference implementation developed by the NOAA/NESDIS/STAR Aerosols and Atmospheric Composition Science Team, our original code did not explicitly account for non-physical solutions of the satellite–Earth intersection geometry (i.e., cases where the quadratic discriminant becomes negative). This omission led to spatial distortions in the derived geolocation fields and, consequently, an incorrect cloud field representation.**

**We have corrected this issue by revising the reprojection procedure to fully enforce the physical validity of the projection geometry and by adopting a numerically robust formulation consistent with the NOAA reference algorithm, including explicit discriminant filtering and proper quadrant handling in the inverse trigonometric calculations (Figure 2). Using this corrected approach, all ABI brightness temperature fields were reprocessed and the corresponding cloud imagery was regenerated.**

[Figure]

**Figure 2. ABI/GOES-16 BT11 at 00:10 UTC on 13 August 2024. Red dots indicate MOD14A1 active fire detections. Blue dots, orange dots, and red crosses denote small, medium, and large maximum fire radiative power (MaxFRP), respectively.**

Comparing this with Figure 4a-b reveals a totally different cloud landscape between the two. Thus it appears that the Figure 4 imagery is in error. I checked imagery from 24 hours earlier and later and did not find anything resembling a match with Figure 4. That raises the question of when the cloud fields in Figure 4 occurred. To the extent that the Figure 4 image and trajectory analysis informs W25's findings and conclusions, it appears that this apparent error should be addressed.

**Response: We thank the Community Commenter for carefully examining Figure 4 and for drawing our attention to the apparent inconsistency in the cloud field depiction. In response to this comment, Figure 4 has been fully replaced with a corrected version (now shown as Figure 1 in this paper). The revised figure is based on a reprocessed and independently verified geolocation of the GOES-16 ABI data and now exhibits cloud structures that are consistent with ABI observations at the stated times. We have also independently verified that the observation times indicated in the revised figure correctly correspond to the ABI scan times reported in the file metadata.**

**All analyses presented in the manuscript have been rechecked using the corrected geolocation. Although the visual appearance of the figure has changed, the physical interpretation of the fire-driven convection, aerosol–cloud interactions, and the associated trajectory-based analysis remains consistent with the revised observations. We therefore confirm that the conclusions of the manuscript are supported by the corrected analysis. We thank the Community Commenter again for identifying this issue, which has led to a substantial improvement in the accuracy and reliability of the presented results.**

Another perceived shortcoming of Figure 4 and attendant discussion is that no fires were identified as the source of pyroCb convection. Given the fact that the WPIE did not document any pyroCbs near the launch points of Figure 4's trajectories, it is suggested that any new source-receptor analysis performed by W25 include a pyroCb image analysis that includes identification of the fire generating the pyroconvection. Lastly, it would be helpful for W25 to use trajectory launch altitudes representative of the cloud heights inferred from the GOES BT imagery. Presently, W25 use 3, 5, and 7 km above ground level, which are low relative to expectations for exhaust from a mature pyroCb.

Response: Following the reviewer's suggestion, we revised the trajectory initialization heights to be consistent with the cloud-top and smoke injection altitudes inferred from GOES-16 satellite observations (Figure c-e). Specifically, forward trajectories were initialized at 4, 8, and 11 km AGL at 23:10 UTC and at 4, 7.5, and 10 km AGL at 23:50 UTC on 13 August.

These heights were selected based on a combination of GOES-16 ABI 11 μm brightness temperature signatures and independent vertical profiling from the EarthCARE mission. CPR radar reflectivity and ATLID lidar backscatter profiles indicate that the fire-driven convective clouds and associated smoke layers extended well into the upper troposphere, with multiple vertically stratified outflow levels. The revised launch altitudes therefore better represent the vertical structure of mature pyroCb exhaust and ensure that the source and receptor analysis is dynamically consistent with the observed cloud depths.

In addition, we have overlaid MOD14A1 active fire detections in the revised Figure 4. Each identified pyroCb cloud feature is spatially collocated with contemporaneous fire hotspots, providing independent confirmation that the observed deep convective clouds are fire-driven rather than meteorologically forced convection. This additional constraint strengthens the robustness of the pyroCb identification and addresses the concerns raised by the reviewer regarding event attribution.

The revised manuscript text addressing this point is provided below for clarity.
[Revised manuscript text:]
"To further assess the vertical structure of smoke-laden cloud systems, we utilized radar reflectivity profiles from the CPR onboard the EarthCARE satellite. As can be seen from Figure 3, CPR overpasses coinciding with maxima in the OMPS UV Aerosol Index (UVAI) enabled a detailed cross-examination of the vertical layering of fire-related clouds and smoke plumes (Fromm et al., 2005; Peterson et al., 2018). Figure 3a illustrates the spatial distribution of the high UVAI region across a large portion of Canada on 13 August. Combined with CPR radar echo measurements from the adjacent overpass periodand the collocated CTH obtained by ATLID

[Figure]

**Figure 3. (Figure 3. in this paper) Satellite Observations of Wildfire Smoke and Vertical Aerosol-Cloud Structure on August 13 and 17, 2024. (a,b) Suomi-NPP/VIIRS true-color imagery overlaid with Suomi-NPP/OMPS UV Aerosol Index on August 13 and 17, 2024, respectively. The black line indicates the EarthCARE satellite ground track, with the CPR reflectivity segment highlighted (note: width not to scale). Red dots indicate all TERRA and AQUA MODIS fire detections for the preceding 24h at native resolution. Panel (a) covers 10:00:25-20:09:22 UTC; panel (b) covers 11:47:50-21:05:06 UTC, both from east to west. (c-e) Radar reflectivity factor**

from CPR/EarthCARE and cloud top height(CTH) of thick clouds from ATLID/EarthCARE along the satellite track on August 13 at 08:53:29 UTC , 21:37:08 UTC and on August 17 at 15:06:57 UTC, respectively, showing the vertical structure of aerosol and cloud. Gray dashed lines show the boundary between the troposphere and the stratosphere. Steelblue dots indicate the Cloud top height from ATLID/EarthCARE.

(Figure 3c-e), the vertical echo profiles reveal canonical features of deep convective cloud systems, characterized by strong radar returns extending through the troposphere. Above and downstream of the convective cores, enhanced ATLID CTH coincide with coherent CPR reflectivity, indicating the presence of optically thin ice clouds in the upper-level outflow region. These elevated, tenuous ice cloud layers are consistent with smoke anvil and plume outflow generated by fire-driven deep convection, highlighting the vertical coupling between intense pyroconvective updrafts and subsequent upper-tropospheric smoke and ice cloud. Figure 3b shows that the high UVAI zones on August 17 were distributed over the wildfire source areas in Canada and the eastern Atlantic region, indicating that wildfire smoke plumes had been transported to the western coastal waters of the European continent. Figure 3e shows the CPR radar echo results, indicating that the cloud body is primarily composed of high-altitude small ice crystals and snow grains, with a lack of low-altitude precipitation echoes. This structure is typical of PyroCb during the decay phase of their transport, when only high-altitude ice crystals remain, between 45°N and 65°N. When this phenomenon aligns with the smoke aerosol transport process, it further indicates that it is a fire-driven PyroCb ice cap remnant.".

References

Peterson, D.A., Berman, M.T., Fromm, M.D. et al. Worldwide inventory reveals the frequency and variability of pyrocumulonimbus and stratospheric smoke plumes during 2013–2023. npj Clim Atmos Sci 8, 325 (2025). https://doi.org/10.1038/s41612-025-01188-5

Response: We thank the reviewer for drawing our attention to the recent global inventory of pyrocumulonimbus events by Peterson et al. (2025). Following this suggestion, we have incorporated this reference into the revised manuscript and explicitly aligned our pyroCb identification strategy with the physically based criteria described therein.

In the revised text, candidate pyroCb features are identified using a multi-criteria framework requiring the simultaneous occurrence of anomalously low 11 μm brightness temperatures indicative of deep convective cloud tops, optically thick cloud signatures inferred from 11-13 μm

brightness temperature differences, enhanced positive 4-11 µm BTD signals, and spatial collocation with contemporaneous MOD14A1 active fire detections. This conservative approach reduces contamination from non-fire-related convection and isolates robust fire-driven convective events, consistent with the methodology and physical interpretation presented in Peterson et al. (2025). The inclusion of this reference strengthens the methodological grounding of the pyroCb identification and places our regional case analysis within the context of recent global-scale assessments.

The revised manuscript text addressing this point is provided below for clarity.

[Revised manuscript text:]

"To characterize fire-driven deep convection and the associated cloud-top properties, we analyzed two complementary thermal infrared indicators from GOES-16/ABI (Figure 4a-d): the BT11, which serves as a proxy for convective cloud-top cooling and vertical development, and the BTD4-11, which has been extensively applied to diagnose interactions between fire activity and deep convection, as well as cloud-top microphysical anomalies associated with pyroCb events (Peterson et al., 2017). This indicator combination has been widely applied to identify smoke-laden convection and cloud-top aerosol signatures in mid-latitude fire regimes (Apke et al., 2018; Fromm et al., 2005).

Candidate pyroCb features were identified using a physically based, multi-criteria framework that required the simultaneous occurrence of anomalously low BT11 values indicative of deep convective cloud tops, optically thick cloud signatures inferred from the BTD11-13, enhanced positive BTD4-11 signals, and spatial collocation with contemporaneous MOD14A1 active fire detections. This conservative methodology reduces contamination from convection unrelated to fire activity and isolates robust convective events driven by wildfires (Peterson et al., 2025). ".

**References**

Apke, J.M., Mecikalski, J.R., Bedka, K., Mccaul, E.W., Homeyer, C.R., Jewett, C.P., 2018. Relationships between deep convection updraft characteristics and satellite-based super rapid scan mesoscale atmospheric motion vector-derived flow. Monthly Weather Review 146, 3461–3480. https://doi.org/10.1175/MWR-D-18-0119.1

Fromm, M., Bevilacqua, R., Servranckx, R., Rosen, J., Thayer, J.P., Herman, J., Larko, D., 2005. Pyro-cumulonimbus injection of smoke to the stratosphere: Observations and impact of a super blowup in northwestern Canada on 3–4 August 1998. Journal of Geophysical Research: Atmospheres 110. https://doi.org/10.1029/2004JD005350

Li, Y., Dykema, J.A., Peterson, D.A., Feng, X., Shen, X., June, N.A., Fromm, M.D., McHardy, T.M., Jacquot, J.L., Pittman, J.V., Daube, B.C., Wofsy, S.C., Dean-Day, J., Rapp, A.D., Bowman, K.P.,

Cziczo, D.J., Mickley, L.J., Pierce, J.R., Keutsch, F.N., 2025. Enhanced radiative cooling by large aerosol particles from wildfire-driven thunderstorms. Sci Adv 11, eadw6526. https://doi.org/10.1126/sciadv.adw6526

Peterson, D.A., Berman, M.T., Fromm, M.D., Servranckx, R., Julstrom, W.J., Hyer, E.J., Campbell, J.R., McHardy, T.M., Lambert, A., 2025. Worldwide inventory reveals the frequency and variability of pyrocumulonimbus and stratospheric smoke plumes during 2013–2023. npj Clim Atmos Sci 8, 325. https://doi.org/10.1038/s41612-025-01188-5

Peterson, D.A., Campbell, J.R., Hyer, E.J., Fromm, M.D., Kablick, G.P., Cossuth, J.H., DeLand, M.T., 2018. Wildfire-driven thunderstorms cause a volcano-like stratospheric injection of smoke. npj Clim Atmos Sci 1, 30. https://doi.org/10.1038/s41612-018-0039-3

Peterson, D.A., Hyer, E.J., Campbell, J.R., Solbrig, J.E., Fromm, M.D., 2017. A Conceptual Model for Development of Intense Pyrocumulonimbus in Western North America. https://doi.org/10.1175/MWR-D-16-0232.1